# Quantifying the Value of Lateral Views
# in Deep Learning for Chest X-rays

**Mohammad Hashir**[1,2*]                                    mohammad.hashir.khan@umontreal.ca
**Hadrien Bertrand**[1*]                                          hadrien.bertrand@mila.quebec
**Joseph Paul Cohen**[1,2]                                         joseph@josephpcohen.com
[1]*Mila, Quebec Artificial Intelligence Institute*
[2]*Université de Montréal*

## Abstract

Most deep learning models in chest X-ray prediction utilize the posteroanterior (PA) view due to the lack of other views available. PadChest is a large-scale chest X-ray dataset that has almost 200 labels and multiple views available. In this work, we use PadChest to explore multiple approaches to merging the PA and lateral views for predicting the radiological labels associated with the X-ray image. We find that different methods of merging the model utilize the lateral view differently. We also find that including the lateral view increases performance for 32 labels in the dataset, while being neutral for the others. The increase in overall performance is comparable to the one obtained by using only the PA view with twice the amount of patients in the training set.

**Keywords:** convolutional neural networks, chest x-rays, lateral views, multi-label classification

## 1. Introduction

Large scale public chest X-ray datasets usually have had only the posteroanterior (PA) view available (e.g. the ChestX-ray14 (Wang et al., 2017) dataset). This has allowed the development of several convolutional neural networks (CNN) based approaches that are built to use only the PA view for automatic prediction (Rajpurkar et al., 2017; Yao et al., 2017; Rajpurkar et al., 2018; Li et al., 2018; Cohen et al., 2019). Other views, such as the lateral (L) view, are not commonly acquired as they are difficult to read without specific training (Feigin, 2010) and only considered useful for specific diagnoses. The lateral view, in particular, is now usually replaced by a CT scan which is only ordered if the PA view is insufficient to diagnose. This practice delays any diagnosis or other actions as the patient would typically need to schedule another appointment. It also increases risk of exposure to larger doses of radiation used in performing CT scans.

There are specific cases in which the lateral view provides information for diagnosis that isn't clear or visible on the PA view (Shiraishi et al., 2007; Feigin, 2010; Ittyachen et al., 2017). For example, up to 15% of the lung can be obscured by cardiovascular structures and the diaphragm (Raoof et al., 2012). It is unclear if the information is completely missing from the PA view, or if it is present but in a way that makes it too difficult for a human

---

[*] Contributed equally

to read. In the first case, the lateral view becomes relevant for making a prediction but a sufficiently advanced model could make it redundant in the second.

This question was previously challenging to answer due to the lack of public large scale datasets of paired PA and L views. But the previous year has seen the release of sizeable de-identified chest X-ray datasets containing multiple views, namely CheXpert (Irvin et al., 2019) & MIMIC-CXR (Johnson et al., 2019) from the United States and PadChest (Bustos et al., 2019) from Spain. This provides us with an opportunity to explore the usefulness of the lateral view. In this work, we analyse the efficacy of multi-view models on the PadChest dataset. Preliminary work by Bertrand et al. (2019) suggests there is value in predicting from a lateral image (with a single view model) for certain radiological labels. We choose PadChest for our analysis due to the variety of radiological labels: it has a total of 194 distinct labels compared to 14 in the other two. This enables a much more fine-grained analysis of how the lateral view can contribute for chest X-ray prediction. We investigate the following questions —

- *Is there a benefit of also including the lateral view in a prediction model? If so, in which cases specifically?*

- *Is there a trade-off between training on PA views of a large amount of patients and training on paired PA & L views of a smaller amount of patients?*

The structure of the paper is as follows. We provide a brief overview of multi-view CNNs in radiology in §2 and a description of PadChest and the preprocessing we used in §3. We present the models we used in §4. The experiments and results are shown in §5, and finally we address these questions with our findings in §6.

## 2. Related work

Using multiple views can help in increasing detection performance in radiology. Setio et al. (2016) achieved an increase of around 54% and 70% in baseline metrics when the number of views was increased to 3 and 9 respectively from 1 in a pulmonary nodule detection task. Shachor et al. (2019) found that their 'Mixture of Views' models achieved higher metrics than single view methods for breast cancer classification and brain MRI segmentation tasks. Multi-view networks also increased performance in tasks like mammogram classification (Geras et al., 2017; Carneiro et al., 2017; Trent Kyono et al., 2019; Nasir Khan et al., 2019), emphysema classification (Bermejo-Peláez et al., 2018), lung segmentation (El-Regaily et al., 2019), fracture detection (Kitamura et al., 2019) and lesion detection (Li et al., 2019).

These works provide a trend of a gain in performance when multiple views are used, which motivates our work. The use of the lateral view in deep learning on chest X-rays has been limited. Rajkomar et al. (2017) tried to predict whether an X-ray was a frontal view or lateral on a dataset from California. Rubin et al. (2018) is one of the initial works that assessed whether combining the frontal and lateral views would help the network. They found that combining the PA and L views led to an increase of 3% in the average AUC over all labels and improved the performance for 12 of the 14 labels in MIMIC-CXR.

There has not been any assessment of multi-view models on the PadChest dataset, to the best of our knowledge. While Yao et al. (2019) performed a preliminary benchmark on

a simple binary classification (normal/abnormal), it used only the PA view. Bertrand et al. (2019) observed that the lateral views did contain useful information for some prediction tasks on PadChest which is the primary motivation for our analysis on this dataset.

## 3. Data and preprocessing

We use the PadChest (Bustos et al., 2019) dataset in our analysis. It is comprised of 160,000 chest X-rays and reports gathered from a Spanish hospital spanning over 67,000 patients with multiple visits and views available. The images have been annotated with various types of radiological findings and differential diagnoses.

We extract the set of patients who have a paired PA and lateral view available forming a total of 30,699 patients. We use only the first study from every patient and discard any additional ones. We resize the images to $224 \times 224$ pixels, utilizing a center crop if the aspect ratio is uneven, and scale the pixel values to $[-1, 1]$ for the training. Each visit can have any number of labels from the total of 194. Since the PadChest dataset defines a hierarchy of labels, we mapped the labels to their respective top level one, in order to maximize the number of images for each label. From those top level labels, we retain only those occurring in at least 50 patients which reduces our set of labels to 64. Some of them are of low clinical interest, such as "electrical device", however they provide a sanity check on the results of the models. The numbers of samples per label we have kept is shown in Figure 3.

## 4. Models

All models are built with four dense blocks following the same configuration as DenseNet-121 (Huang et al., 2017; Rajpurkar et al., 2017). Along with one type of single view model, we test four different methods of combining both views.

1. **DenseNet**: This is a standard DenseNet-121 with its fully connected layer modified for a multi-label output. It is used as a single view model and serves as the baseline. We use two DenseNets, each trained and tested on the PA and L views respectively.

2. **Stacked**: The architecture is essentially the same as a standard DenseNet-121 but the lateral view is stacked as a second channel of the PA image. It is the simplest way to combine the views before giving them to a model. This model is not designed to work if only one view is available.

3. **DualNet** (Rubin et al., 2018): The PA and L views are processed by two separate CNNs composed of four DenseNet blocks each. Their output feature maps are then passed through a global average pooling layer, concatenated and given to a fully connected layer that maps them to the output labels. Similar to Stacked, this model also relies on both views to be available to make a prediction.

4. **HeMIS** (Havaei et al., 2016): The HeMIS architecture involves propagating each input view/modality through its separate set of layers, combining them by calculating pixelwise statistics and propagating further for classification. We create a HeMIS-style architecture where the PA and L views are processed separately by two separate CNNs 'branches' composed of the first three DenseNet blocks each. The mean and variance

of the output feature maps of both CNNs are computed pixelwise which are then concatenated and given to another dense block. The output of this block is passed through a global average pooling layer and given to a fully connected layer that maps them to the output classes. This model works even if only one view is available, with variance of the feature maps substituted with a tensor of zeroes of the same shape.

5. **AuxLoss**: We modify DualNet by adding two separate fully connected layers to the PA and L branches respectively that map it to the same output labels and train all three losses jointly. Other than concatenating the pooled vectors of the branches, we also add an option to average them. Inspired by Lee et al. (2015) (we classify the output of each branch rather than layer) and Trent Kyono et al. (2019), this could be considered as a DualNet regularized by auxiliary view-specific losses. This model is designed to be robust to missing views; if a view is not present the prediction is calculated off the available view's fully connected layer.

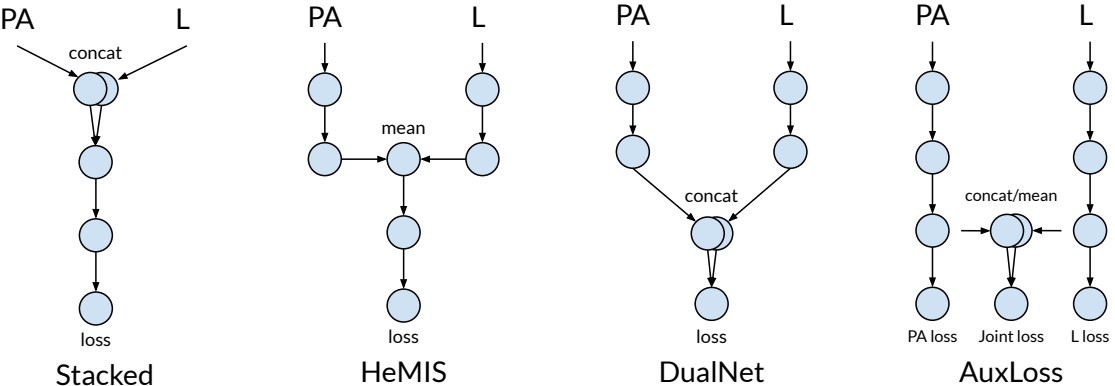

Figure 1: High level topology of the multi-view models used. PA and L indicate the input images while "loss" indicates where the prediction error is computed. Arrows indicate the computation flow during the forward pass.

## 5. Experiments

We train all models for 40 epochs with Adam and a batch size of 8. Other hyperparameters differ for the models. In addition, we use curriculum learning with HeMIS and AuxLoss denoted by the '-CL' suffix hereon. A description of our training details including hyperparameters and how we implement curriculum learning is given in Appendix B.

### 5.1. Performance

We evaluate the multi-view models using the AUC (averaged over the 64 labels) achieved in three configurations of the test set. The first is the general case where paired views are available for all patients. The other two are the cases when only a single view is available:

either only the PA view is present or the L view. For models that require both views to be available (Stacked and DualNet), we use a zero matrix in place of the missing view. With this testing regime, we want to see how the multi-view models perform with only one view relative to the respective single view model. We perform five runs of every model with a different random split every time and report the results, averaged over all runs, in Table 1.

It is apparent that there is very little difference between the multi-view models and most of the models' confidence intervals overlap when both views are available. We perform t-tests between pairs of all models' performances and find that only two pairs have differences that are statistically significant. AuxLoss-CL achieves a significantly better performance than DualNet ($p = 0.003$) and Stacked ($p = 0.017$) but not the rest. A very interesting observation is that the Stacked model performs as well as the rest, despite the lack of a spatial correspondence between the channels of the input.

The multi-view models diverge in performance when given only a single view. In the case of testing on only PA images of patients in the test set, DualNet emerges as a clear winner. Its confidence interval seems very close to that of Stacked, AuxLoss and AuxLoss-CL but a t-test proves the difference is statistically significant ($p = 0.012$, $p = 0.002$, $p = 0.001$ respectively). The AUC of the single view DenseNet-PA lies between DualNet and Stacked and we find that the two multi-view models are not significantly better or worse respectively. Curriculum learning also seems to not have a discernible effect, in fact HeMIS with curriculum learning is much worse than a vanilla HeMIS. When evaluating on only lateral images, there is a large change in the performance. While the AuxLoss models experience a 5-7% decrease in AUC relative to paired images, the other models deteriorate by more than 20%. The single view model is better than the multi-view in this case.

Table 1: Test AUC achieved by the different models, averaged over five runs with standard deviation also reported. DenseNet is trained on a single view denoted by the suffix, rest are trained on both views. The symbol in the superscript indicates the difference between a pair of models in the same column is statistically significant.

| Model | Test AUC | | |
|---|---|---|---|
| | Both | PA | L |
| DenseNet-L | — | — | $0.780 \pm 0.004$ |
| DenseNet-PA | — | $0.793 \pm 0.007$ | — |
| Stacked | $0.804 \pm 0.003^*$ | $0.786 \pm 0.009^\ddagger$ | $0.595 \pm 0.046$ |
| DualNet | $0.801 \pm 0.003^\dagger$ | $\mathbf{0.800 \pm 0.004}^{\dagger * \ddagger}$ | $0.539 \pm 0.018$ |
| HeMIS | $0.803 \pm 0.006$ | $0.758 \pm 0.014$ | $0.603 \pm 0.044$ |
| HeMIS-CL | $0.803 \pm 0.007$ | $0.723 \pm 0.017$ | $0.627 \pm 0.036$ |
| AuxLoss | $0.803 \pm 0.006$ | $0.787 \pm 0.005^*$ | $0.753 \pm 0.002$ |
| AuxLoss-CL | $\mathbf{0.809 \pm 0.003}^{*\dagger}$ | $0.788 \pm 0.005^\dagger$ | $\mathbf{0.771 \pm 0.003}$ |

### 5.2. How does the lateral view help in prediction?

As evidenced in Section 5.1, there is a major difference between the multi-view models when it comes to the view(s) they have to use for making a prediction. We wanted to examine how the AUC is affected when the proportion of patients having a paired L view in the test set is varied. We do this in Figure 2 by iteratively removing the lateral view for 1% of the patients and testing the models, incrementing the percent removed until 100%.

The most conspicuous observation is that the DualNet does not seem to use the lateral view for prediction at all. HeMIS is also unusual where adding paired views actually hurts performance until the proportion crosses a threshold. This could be caused by the usage of variance in combining the views: when HeMIS was tested on only PA images, the variance of the feature maps was substituted with a zero tensor causing the model to believe there was zero variance. As we add patients with paired lateral views, smaller proportions of paired views led to high variance causing the dip in the curve in Figure 2. Past a certain proportion, the variance of the feature maps started approximating the population variance which led to increasing performance.

The performance of the multi-view models in the three testing regimes (Table 1) suggests that the AuxLoss model combined with curriculum learning utilizes the lateral view most productively. We use AuxLoss-CL to find the effect of the lateral view on individual labels in the dataset: we perform a comparison between the label-wise AUC of AuxLoss-CL and DenseNet-PA averaged over all runs. If the difference between the average AUC for a given label is lesser than the standard deviation of the better model, we annotate that label as indifferent to the model used. We find that 32 of the 64 labels see an improvement in AUC when both the views are used jointly, visualized in Figure 3

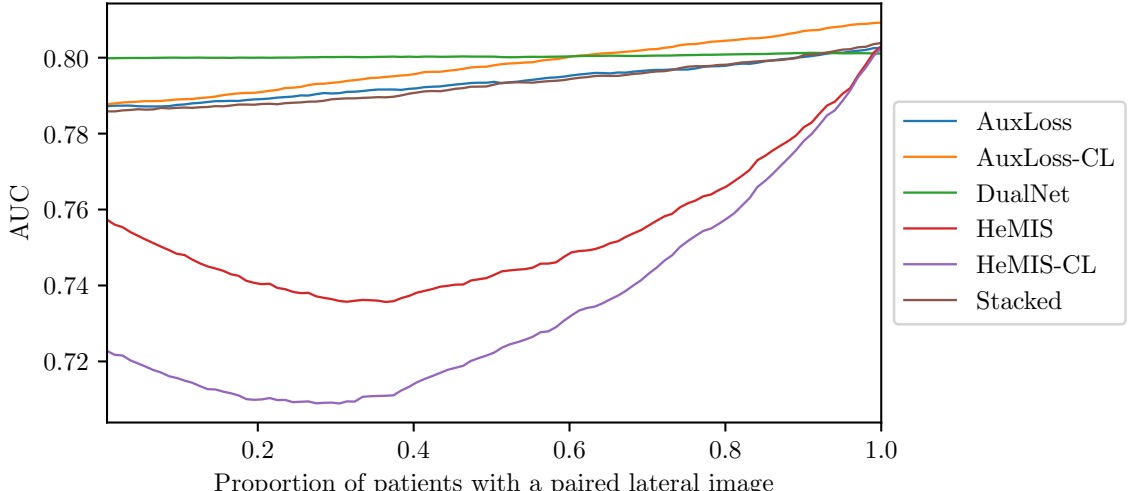

Figure 2: Change in AUC as proportion of patients with paired lateral views increase. Only multi-view models can be evaluated here. Here 0.2 indicates that 20% of the PA image samples used to test the model had a corresponding L view and the rest were treated as single images (If they had a L view in the dataset it was ignored).

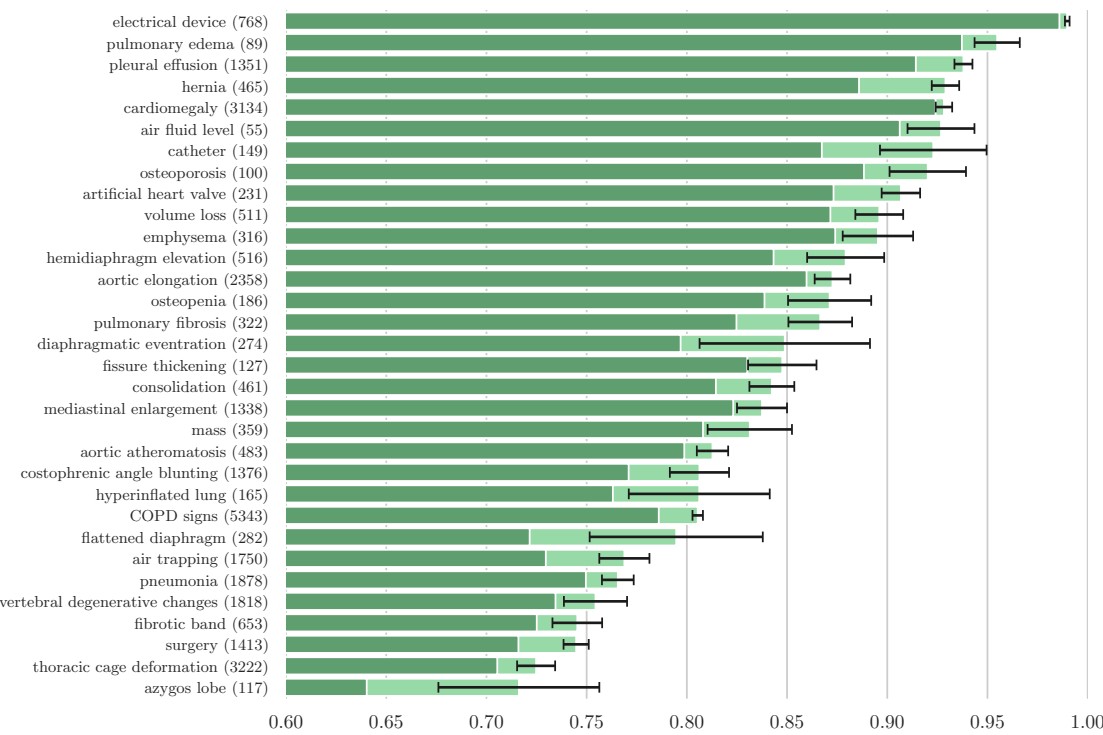

Figure 3: The subset of the 64 labels we use in PadChest that see an improvement in AUC with AuxLoss-CL. The improvement is considered relative to the AUC achieved by the single view DenseNet-PA which is denoted by the darker bar. The number in the brackets refers to the number of samples with the label

## 5.3. Benefits of increasing the training set size

We form a second dataset by adding the remaining patients that have a PA view available; we call this the **extended** dataset. This adds 27,576 PA images from new patients which are used only for training the model. We trained the DenseNet-PA and AuxLoss-CL models on this extended dataset, and evaluated them on the main test set. Those results are reported in Table 2.

The DenseNet-PA model gains one percent of AUC from doubling the dataset size. This marginal increase in performance from a major increase in dataset size is consistent with what is observed in other computer vision tasks (Sun et al., 2017). We examine how the AUC of individual labels changes between the DenseNet-PA trained on the main and extended datasets and find that, similar to AuxLoss-CL, 32 labels see an increase with 22 of them overlapping with the AuxLoss-CL. The top three increases are in the labels 'azygos lobe' (+24.1%), 'tracheal shift' (+12.4%) and 'diaphragmatic eventration' (+6.3%). A plot detailing all the improvements in the style of Figure 3 is included in Appendix A.

AuxLoss-CL is indistinguishable in performance from the extended DenseNet-PA. It also became worse at utilizing both views, likely due to the dominant presence of PA images.

Careful over-sampling of joint and lateral images during the curriculum learning might fix that problem.

Table 2: AUC and standard deviation of the DenseNet-PA and AuxLoss models trained on the main and extended dataset but evaluated on the same main test set. The size of the training set for the extended dataset is double that of the main. The Main AUC column copies the values from Table 1 for easier reference.

|  |  | Train Data (AUC on test reported) | |
| Model | Test Data | Main | Extended |
| --- | --- | --- | --- |
| DenseNet-PA | PA | $0.793 \pm 0.007$ | $0.813 \pm 0.005$ |
| AuxLoss-CL | PA | $0.788 \pm 0.005$ | $0.812 \pm 0.006$ |
| AuxLoss-CL | Both | $0.809 \pm 0.003$ | $0.772 \pm 0.018$ |

## 6. Discussion

To return to the questions posed in the introduction, it appears a well-tuned PA-only model is competitive with a well-tuned joint model. However, for the specific labels shown in Figure 3, the lateral view allows a statistically significant improvement relative to using just the PA view. We proposed a joint model using auxiliary tasks that gives strong results with any subset of views.

While all multi-view models offer similar performance when given both views, they do not perform as well when given only one view. Testing these models on different proportions of paired L images brings very intriguing observations to surface. First, it indicates that models other than AuxLoss were relying too heavily on the PA view to make a prediction, especially DualNet which completely ignores the L view. Second, it might explain how the Stacked model performs well on the first two testing regimes: it just learns to use less information from the second channel of the input image to favour predicting from the first channel only. Third, adding curriculum learning to AuxLoss mitigates this reliance on the PA view but does not have much effect on HeMIS.

For the second question, we found that doubling the number of PA images in the training set only gives a marginal increase in performance. A joint model trained on only half the data gives the same performance. If it is more costly to bring in a new patient than to acquire another view for an already present patient, then our results suggest acquiring both views for a smaller number of patients rather than one view on twice that number.

We also find that although the different approaches achieve similar performance with both views, the training is less sensitive to hyperparameters for the AuxLoss and DualNet models shown in Figure 4. With this result, we would conclude that these models would be easier to train as the range of optimal hyperparameters would be relatively wider.

This study has two limitations. First, while we have shown that the lateral view is useful for some labels, we did not sort those by clinical relevance. This can be seen notably by the presence of "electrical device" or "catheter". It might be that most of those labels

are extremely rare, or typically detected through other means than a chest x-ray. Second, this study is based entirely on the PadChest dataset. The population differs from other available datasets, at least by ethnicity, but likely also by clinical practice. In other words, we expect both covariate shift and concept shift (Moreno-Torres et al., 2012). This implies that while the lateral view will still be useful for many labels, those labels might not be the same as the ones we found here.

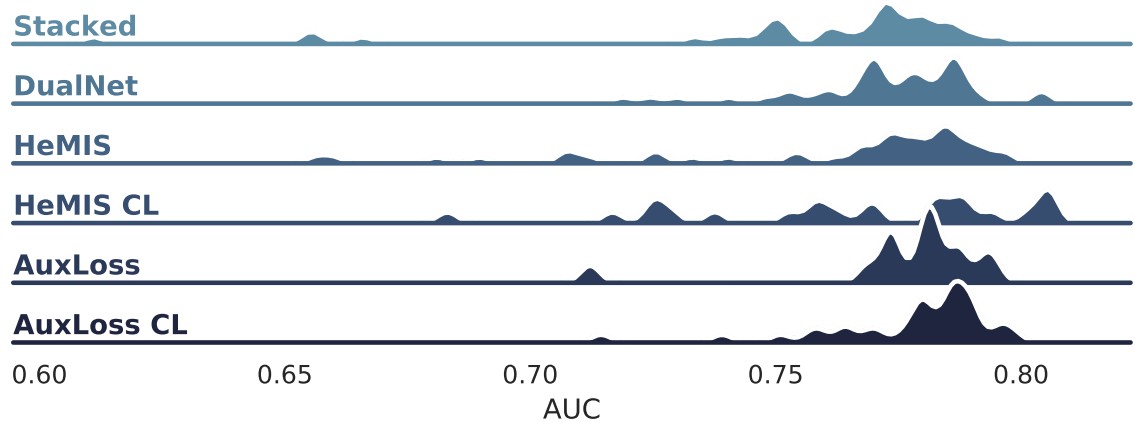

Figure 4: Distributions of AUC for a 40 combination hyperparameter search for each model. Some models are much more robust to hyperparameter changes than others.

## Acknowledgments

We thank Paul Bertin for his code and Mathieu Germain for discussions. We thank AcademicTorrents.com for making data available for our research. This work is partially funded by a grant from the Institut de valorisation des donnees (IVADO). This work utilized the supercomputing facilities managed by Mila, NSERC, Compute Canada, and Calcul Quebec. We also thank NVIDIA for donating a DGX-1 computer used in this work.

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

# Appendix A. Improvement in labels with extended dataset and paired views

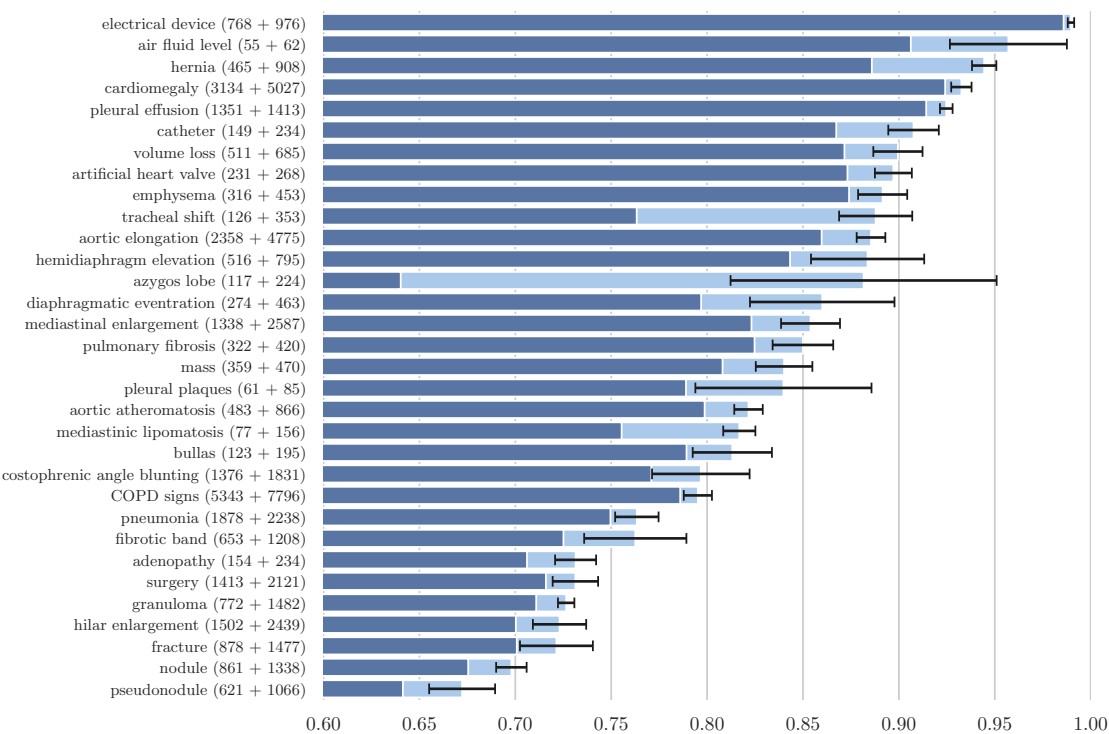

Figure 5: The subset of the 64 labels we use in PadChest that see an improvement in AUC when the **extended dataset** is used with the DenseNet-PA model. The improvement is considered relative to the AUC achieved by the single view DenseNet-PA on the main dataset which is denoted by the darker bar. The number in the brackets before and after the plus sign refers to the number of samples in the main and extended dataset with the label

Table 3: Increase in AUC relative to DenseNet-PA (Main) for labels that improved with both DenseNet-PA (Extended) and AuxLoss (Main). Averaged over all runs

| Labels | AUC increase | |
| --- | --- | --- |
| | DenseNet-PA (Extended) | AuxLoss-CL (Main) |
| azygos lobe | 0.241 | 0.076 |
| diaphragmatic eventration | 0.063 | 0.052 |
| hernia | 0.059 | 0.043 |
| air fluid level | 0.051 | 0.021 |
| hemidiaphragm elevation | 0.040 | 0.036 |
| catheter | 0.040 | 0.056 |
| fibrotic band | 0.037 | 0.020 |
| mass | 0.032 | 0.023 |
| mediastinal enlargement | 0.031 | 0.014 |
| volume loss | 0.028 | 0.024 |
| costophrenic angle blunting | 0.026 | 0.035 |
| aortic elongation | 0.026 | 0.013 |
| pulmonary fibrosis | 0.025 | 0.042 |
| artificial heart valve | 0.024 | 0.034 |
| aortic atheromatosis | 0.023 | 0.014 |
| emphysema | 0.018 | 0.021 |
| surgery | 0.015 | 0.029 |
| pneumonia | 0.014 | 0.016 |
| pleural effusion | 0.010 | 0.024 |
| COPD signs | 0.009 | 0.019 |
| cardiomegaly | 0.008 | 0.004 |
| electrical device | 0.004 | 0.004 |

## Appendix B. Hyperparameters

### B.1. Stability of training

For each model, we performed a random search using Orion (Bouthillier et al., 2019) over the learning rates, the amount of dropout, and for CL models the probability of dropping a view for each batch. 40 combinations of those hyperparameters were tried per model.

In Figure 4, we show the distributions of the AUC on the validation set over the combinations, per model. We observe strong variations in the shape of those distributions. HeMIS, for example, is very sensitive to the choice of hyperparameters, whereas AuxLoss is much more concentrated.

### B.2. Training details

For all models, we use the Adam optimizer and a batch size of 8 and train up to 40 epochs. We also compute class weights (clamped at 5) to balance the loss as the labels are highly imbalanced. All models use early stopping based on the AUC achieved on the validation set.

**Learning rate**   The initial learning rate differs for every model but it is decayed by a factor of 10 halfway through training for all. These model-specific hyperparameters were found through an extensive random search and are given in Table 4.

**Dropout**   We use dropout with a different probability for each model, given in Table 4

**Dataset**   We utilize a 60-20-20 split in the dataset for train, validation and test sets. We also use data augmentation such as adding random translations and rotations. We also add random Gaussian noise to the image.

**Loss weights for AuxLoss**   AuxLoss uses a weighted sum of the three losses (averaged over all labels) with weights of 1.0, 0.3 and 0.3 for the joint, PA and L losses.

**Curriculum learning**   On HeMIS, we randomly drop one of the views with probability 0.25 each. For AuxLoss, we randomly select one of the PA or L **loss** with probability of 0.2 for each and use that to update that view's branch instead of the entire model. The weighted sum of the three losses is used to update the entire model only 60% of the time. This is done to make the model rely less on having both views available all the time.

Table 4: Hyperparameters of the best models found. Most joint models are composed of 3 parts, each with a different learning rate: the PA branch, the L branch, and a common branch. Curriculum learning (CL) models have an additional hyperparameter which is the probability of dropping one view for any given sample.

| Model | LR | Dropout | View dropping |
|---|---|---|---|
| DenseNet-PA | $5.8e^{-4}$ | 0.0 | |
| DenseNet-L | $2.6e^{-4}$ | 0.2 | |
| Stacked | $1.9e^{-4}$ | 0.1 | |
| DualNet | $3.0e^{-4}$, $7.6e^{-4}$, $2.7e^{-4}$ | 0.2 | |
| HeMIS | $3.8e^{-4}$, $2.0e^{-5}$, $2.8e^{-5}$ | 0.1 | |
| HeMIS-CL | $1.7e^{-4}$, $5.6e^{-4}$, $7.2e^{-5}$ | 0.1 | 0.5 |
| AuxLoss | $2.1e^{-4}$, $1.9e^{-4}$, $6.6e^{-4}$ | 0.2 | |
| AuxLoss-CL | $6.9e^{-5}$, $9.5e^{-5}$, $5.2e^{-5}$ | 0.1 | 0.4 |

