# OpenReview forum: "Quantifying the Value of Lateral Views in Deep Learning for Chest X-rays"
_MIDL.io/2020/Conference — MIDL 2020_

### Official Review · AnonReviewer2 · 2020-03-09
**Extensive empirical evaluation but limited novelty**

**Rating:** 3
**Confidence:** 3
**Recommendation:** Poster

**Summary:**

The authors consider the problem: are the lateral view images as important as PA view images in chest X-ray datasets for deep learning models. The authors performed extensive comparisons with several different models. The authors conclude that " it appears a well-tuned PA-only model is competitive with a well-tuned joint model".

**Strengths:**

Pros:
This paper performs a detailed comparison between different models, and the ablation study is also complete and convincing. The authors carefully designed experiments to determine the effect of images from two views, and plotted the curve of AUC varying with the proportion of paired lateral images.

**Weaknesses:**

Cons:
This work appears to be limited to empirical evaluations and comparisons to me. Although the authors proposed an "Auxloss CL" model, which is not in the literature. But the idea of combing several different losses (or termed auxiliary loss) has long been existing in the literature such as image segmentation (e.g. PSPNet, Context Encoding, and some earlier works).

**Justification Of Rating:**

Overall, I think the empirical evaluation is quite extensive and solid, but the novelty is limited. However, I'm not sure if such novelty is a hard requirement for MIDL. Therefore, I tend to give a weak-accept rating.

**Paper Type:**

validation/application paper

**Questions To Address In The Rebuttal:**

Could the authors clarify what is the novelty, besides the empirical evaluation of existing models?

**Special Issue:**

no

---

> ### Author Response · Authors · 2020-03-27
> **Thank you for your review**
>
> In a broader context, our novelty is a fine-grained analysis of how including the lateral chest X-ray view can affect the prediction for a *large* set of radiological labels.
>
> About the novelty in the model specifically, we do agree that using auxiliary losses is not unheard of in computer vision and therefore we did not focus too much on that as a contribution. However, we believe that the majority of the use cases of such losses has been to regularize the model from learning too specific features (in most cases, from a single view).
>
> Our novelty mainly lies in how we use these losses to equip a multiview model with a functionality that makes it more robust to missing views, in addition to regularizing feature construction from multiple views. Further by adding curriculum learning, we ensure that the auxiliary losses actually contribute to feature construction rather than being dominated by the joint loss.

---

### Official Review · AnonReviewer1 · 2020-03-13
**Good experimentation. Little novelty. Weak presentation.**

**Rating:** 2
**Confidence:** 5

**Summary:**

The authors investigate if the use of lateral (L) views increase classification performance in chest x-ray diagnoses with respect to the use of only posterior-anterior (PA) views. The database used is the large PadChest database, with a pre-selection of cases that have both PA and L views. Several network architectures are investigated, from a stacked densenet to a two-path convolutional network with a joint auxiliary loss. Further experimentation comparing the use of PA and L views with using twice the length of the dataset but with only PA images is performed. Results show that:
-	When using the same number of training points, the use of both PA and L increase the performance irrespectively of the network employed
-	When training with twice the data, the performance increases considerably
-	When training on PA and using PA and L views to classify the performance decreases


**Strengths:**

Experimental validation. Several networks analyzed with repeats. Meta-parameter stability analysis. Two datasets – posterior-anterior and lateral views and extended posterior-anterior views. Good references, although the reference for PadChest is not complete.

**Weaknesses:**

Experimental paper with a low amount of novelty. Of course, adding more views increase diagnosis performance. That is not any surprise. Adding more data also increase performance. No surprise either.
Is it better to add more views or more data of the same view? According to their data, DenseNet-PA on the extended dataset obtains the best performance metric. It only requires a larger training data. From a patient perspective, acquiring a single PA view is better than acquiring two views, since it involves lower radiation. The conclusion of this paper to me is that, if there is a system to diagnose disease from chest x-rays, the best option would be to train it with a database that is as large as possible with only PA views. Such is not mentioned in the discussion.


**Detailed Comments:**

Lack of clarity around many important aspects on the paper.
- Introduction. The authors refer that the acquisition of CT images delays prognosis. It would delay diagnosis. Please use the right terminology.
- Outcomes. The authors are vague around the selection of outcomes in section 3. It is only on the results section that we learn that there are 64 outcomes selected.
- Typo on section 3 on the range [-1, 1]. Also, how was that scaling performed? Two arbitrary values? Please clarify.
- What is the split of the data in train / validation / test?
- Performance metric. The authors refer continuously to the AUC as performance metric. Is that the average AUC over the 64 measurements? I assume that is the case, but it is unclear from the text.
- 5.1 – performance. Over what variables are the t-test performed? Is it over the means of the per-outcome AUC? Are they gaussian?
- Discussion ‘Doubling the number of PA images in the training set only gives a marginal increase in performance’. For Densenet-PA, the increase if between 0.793 to 0.813, which is the highest AUC obtained overall. I would not consider such marginal.
- Discussion: “‘catheter’ may be diagnosed through other means“. That sentence is nonsensical. A catheter is not diagnosed.


**Justification Of Rating:**

The experimentation is solid but it leads us to expected conclusions. Missing clarity. There is a general lack of interest from the conclusions of the paper, which I consider not to be complete. The authors should think if they are trying to solve a medical need (such as the automated diagnosis) or to address the preference of using extended data to an extended dataset with a subset of the data.

**Paper Type:**

validation/application paper

**Special Issue:**

no

---

> ### Author Response · Authors · 2020-03-27
> **Thank you for your review**
>
> With this work, we wanted to present a deep delve into the role of specifically the lateral view in multi-view chest X-ray (CXR) prediction, an area which relatively has not been explored as much. Our primary goal was to assess how inclusion of the lateral view affected different radiological labels and provide a modeling mechanism that can be robust to a missing PA or L view. We were not aiming to surprise or provide unexpected conclusions; we definitely expected performance to increase with additional views or patients, given most multi-view research in radiology. But real life conditions are data scarce and there is a trade off between more views of the same patient vs. more patients that deep learning researchers/practitioners will need to address. We intend for this paper to aid those people in making decisions about such trade-offs subjective to their conditions. In addition to being a benchmark of multiview models for CXR prediction in research, we also wanted this paper to be a resource for those implementing CXR prediction systems.
>
> > “Typo on section 3 on the range [-1, 1]. Also, how was that scaling performed? Two arbitrary values?”
>
> The pixel values of the images in the dataset lie in [0, 65536]. We rescale the image such that the values lie in [-1, 1] using the formula  2 * (X / 65536) - 1 where X is the pixel value. We chose these values because it is standard practice in deep learning for computer vision to rescale the input to usually [0, 1] or [-1, 1].
>
> > “What is the split of the data in train / validation / test?”
>
> It is 60-20-20. More specifics on the training is given in Appendix B
>
> > “Is that the average AUC over the 64 measurements?“
>
> Yes, it is the mean of the AUC of the 64 labels
>
> > “Over what variables are the t-test performed? Is it over the means of the per-outcome AUC? Are they gaussian?”
>
> Yes. The t-test is performed over the average AUCs obtained in each run over 5 runs of random splits.
>
> >“Discussion ‘Doubling the number of PA images in the training set only gives a marginal increase in performance’. For Densenet-PA, the increase is between 0.793 to 0.813, which is the highest AUC obtained overall. I would not consider it so marginal.”
>
> We meant marginal relative to the number of extra images used in training. What we were trying to say here is that we do not obtain a very substantial increase in AUC when the training set is doubled. Incremental would be a more appropriate word in this context, we shall edit that.
>
> > “The conclusion of this paper to me is  ……. not mentioned in the discussion.”
>
> We chose not to conclude in that manner because having double the number of PA images did not achieve an AUC that was significantly better. The AUC intervals of AuxLoss-CL (Main) and Densenet-PA (Extended) overlap in Table 2: [0.806, 0.812]  and [0.808, 0.818] respectively. We try to leave it as open-ended as possible as we want the reader to form their own takeaways from the paper based on their specific situation.
>
> >“There is a general lack of interest from the conclusions of the paper......to an extended dataset with a subset of the data.”
>
> While we chose to not conclude decisively towards either more paired views or more patients, we can still say that:
>
> - Overall, using a multiview model is definitely better than using only a single view.
> - Different multiview models utilize the lateral view differently. AuxLoss-CL is the most advantageous as it utilizes both views well and is more robust to missing views and changes in hyperparameters.
> - Including the lateral view improves the performance of 32 labels out of the total 64.
> - Training a single view model with double the training samples does not significantly improve the overall average performance over a multiview model trained on half the samples. But it does improve the individual performance of 32 labels, of which 22 overlap with the 32 improved by including the lateral view
>
> If you think that the paper would benefit from a section enumerating the above conclusions along with the advantages of the AuxLoss-CL  model over other multi view models, we could definitely add that to the final version of the paper to add more clarity.
>
> Thank you for your other comments, we shall edit the paper to reflect them.

---

### Official Review · AnonReviewer4 · 2020-03-14
**Review of Quantifying the Value of Lateral Views in Deep Learning for Chest X-rays**

**Rating:** 3
**Confidence:** 4
**Recommendation:** Poster

**Summary:**

The author proposed and compared a number of model architectures for incorporating the lateral view information with the posteroanterior features. Experiments in the paper show there is considerable improvement by adding the lateral view. It also suggests the auxiliary loss topology (with curriculum learning) is a better approach than other concatenation methods.

**Strengths:**

Sufficient experiments and convincing results.
The auxiliary loss structure for combining PA and L is interesting and seems to have a good improvement.
Quantitative comparison for the improvement gain from L over different architectures


**Weaknesses:**

Not much novel modifications for the proposed AuxLoss.
Performance gain seems just from curriculum learning and performance of all PA+L combined architectures without CL seem roughly the same. Need more evidence to support the AuxLoss.


**Justification Of Rating:**

For the sufficient experiments and detailed analysis of the results, I will vote the accept. Methodology may be just incremental but still have some interesting insights. Overall the quality of paper worths a presentation in the conference.

**Paper Type:**

validation/application paper

**Questions To Address In The Rebuttal:**

What was the curriculum learning strategy in this paper?
Did the author try different CL strategies?

**Special Issue:**

yes

---

> ### Author Response · Authors · 2020-03-27
> **Thank you for your review**
>
> > “Performance gain seems just from curriculum learning”
>
> We were unable to conclude this due to how curriculum learning (CL)  behaved differently in HeMIS and AuxLoss. Figure 2 shows conflicting results about the effect of CL on learning: it degrades performance in HeMIS but improves it in AuxLoss.
>
> > “...performance of all PA+L combined architectures without CL seem roughly the same. Need more evidence to support the AuxLoss.”
>
> We do agree that the performance of all PA+L combined architectures without curriculum learning (CL) seem the same. But we’d like to reiterate that this seems applicable to only the “Both” test set (where both views are available for all samples). On testing the multi-view models on only a single view, either the PA or L, they diverge in performance significantly. Table 1 demonstrates that all multiview models except AuxLoss rely heavily on the PA view to make their prediction. These models will be unreliable in real life situations where only the L view happens to be available for input to the model. In this case of having only a single view, AuxLoss-CL is the best multi-view model. We believe that the following is sufficient evidence for AuxLoss-CL over other multiview models
>
> (i)  it has the lowest variance across the three test sets for the multiview models
> (ii) it is more robust to missing views
> (iii) it actually utilizes the lateral view as evidenced in Fig 2
> (iv) it is relatively less sensitive to changes in hyperparameters (Fig 4)
>
> > “What was the curriculum learning strategy in this paper? Did the author try different CL strategies?”
>
> Our curriculum learning is detailed in Appendix B.2. We wanted to save space in the paper and opted to move this and other training specifics to supplementary material. If you think this is important enough to be moved to the main paper, we are open to that. The strategy is given below:
> ‘On HeMIS, we randomly drop one of the views with probability 0.25 each. For AuxLoss, we randomly select one of the PA or L loss with probability of 0.2 for each and use that to update that view's branch instead of the entire model. The weighted sum of the three losses is used to update the entire model only 60% of the time. This is done to make the model rely less on having both views available all the time.’

---

### Meta-Review · Area_Chair1 · 2020-04-07
**MetaReview of Paper35 by AreaChair1**

**Rating:** 3
**Recommendation For Accepted Papers:** Poster

**Metareview:**

There seems to be general consensus that this paper presents an interesting study but with very limited methodological contribution.

**Paper Type:**

validation/application paper

**Special Issue:**

no

---

### Decision · Program_Chairs · 2020-04-11

Accept